# HBcompare: Classifying Ligand Binding Preferences with Hydrogen Bond Topology

**DOI:** 10.3390/biom12111589

**Published:** 2022-10-28

**Authors:** Justin Z. Tam, Zhaoming Kong, Omar Ahmed, Lifang He, Brian Y. Chen

**Affiliations:** Department Computer Science and Engineering, Lehigh University, 113 Research Drive, Bethlehem, PA 19004, USA

**Keywords:** structural bioinformatics, function annotation, specificity annotation

## Abstract

This paper presents HBcompare, a method that classifies protein structures according to ligand binding preference categories by analyzing hydrogen bond topology. HBcompare excludes other characteristics of protein structure so that, in the event of accurate classification, it can implicate the involvement of hydrogen bonds in selective binding. This approach contrasts from methods that represent many aspects of protein structure because holistic representations cannot associate classification with just one characteristic. To our knowledge, HBcompare is the first technique with this capability. On five datasets of proteins that catalyze similar reactions with different preferred ligands, HBcompare correctly categorized proteins with similar ligand binding preferences 89.5% of the time. Using only hydrogen bond topology, classification accuracy with HBcompare surpassed standard structure-based comparison algorithms that use atomic coordinates. As a tool for implicating the role of hydrogen bonds in protein function categories, HBcompare represents a first step towards the automatic explanation of biochemical mechanisms.

## 1. Introduction

Exploring the space of protein structures with algorithms that compare molecular shape can reveal structural similarities that point to shared evolutionary origins and biological functions. The nature of these observations is influenced strongly by the way in which molecular structure is represented. Algorithms that represent protein structure as a geometric arrangement of secondary structure elements [1,2] or as a collection of alpha carbon coordinates [3,4] can reveal relationships between families of protein folds [5,6]. Comparisons of binding sites, represented as collections of atomic coordinates [7,8], molecular surface patches [9,10] or volumetric constructs [11], can identify proteins with similar catalytic functions and different overall folds [12]. Representing binding site geometry or electrostatic isopotentials as geometric solids can reveal differences in binding site geometry and charge that identify mechanisms that alter binding specificity [13,14,15,16].

Existing representations integrate many aspects of protein structure, but none to our knowledge focus exclusively on the topology of hydrogen bonds. Yet hydrogen bonds play a central role in organizing tertiary structure and in governing the specificity of molecular recognition. For this reason, we hypothesize that the topology of hydrogen bonds, alone, can distinguish proteins with different binding preferences, even if they have the same overall fold. To evaluate this hypothesis, we developed *HBcompare*, a deep learning algorithm for comparing the topology of hydrogen bonds in protein structures.

The specific problem studied here begins with a superfamily of proteins that perform the same catalytic function, which have been classified into subfamilies with different binding preferences. The goal is to classify a new protein into one of these subfamilies based on similarities in hydrogen bond topology. In such cases, the superfamily exhibits the same overall fold, so the topology of their hydrogen bonds is largely conserved. At the same time, critical variations in hydrogen bonding patterns could lead to differences in binding specificity that differentiate subfamilies in terms of preferred binding partners. Correctly classifying a protein into one of the subfamilies requires a look beyond the shared similarities of the superfamily to focus on differences that betray subfamily membership.

HBcompare describes the topology of hydrogen bonds in a protein structure using a *molecular graph*, which we define in detail below. As a representation of protein structures, graphs have been used frequently to describe spatial relationships between atoms, amino acids and secondary structure elements (e.g., [17]) or protein structure prediction (e.g., [18]). Rather than represent more aspects of protein structure, HBcompare is first to use graphs that exclusively represent the topology of hydrogen bonds.

This exclusivity enables a novel capability: Since HBcompare atomistically considers only hydrogen bond topology, the classification of a protein into a subfamily with specific binding preferences is also predicting a role for hydrogen bond topology in the specificity mechanism. That is, since only hydrogen bond topology is considered, it must be at least related to the difference between categories. We call this feature "mechanism prediction", and it cannot be performed with holistic methods. In the holistic case, multiple biophysical mechanisms, such as atomic coordinates and electrostatic potentials, are used together in a weighted fashion to distinguish between specificity categories. In such cases, a single mechanism cannot be said to explain the distinction between categories.

The atomistic approach has useful applications. By suggesting a role for hydrogen bonding, HBcompare generates explanations that a non-computational user can adapt into experimental design. For example, if similarities in hydrogen bond topology justify the classification of a protein structure into a category with well defined binding preferences, then it is logical that experiments that mutate hydrogen bond donors and acceptors may reveal the bonds that play an important role in recognition. Without that observation, a much larger space of experimental redesigns must be considered.

Naturally, HBcompare is only a first step in creating possibilities for automatically explaining binding mechanisms. Furthermore, a complete explanation may not always possible, because some biophysical phenomena will co-occur with hydrogen bonds. For example, a protein that lacks one side of a salt bridge differs from one with a complete salt bridge because it might lack a hydrogen bond donor or because it might lack a charged amino acid. We see HBcompare as one tool in an *Analytic Ensemble* that would eventually be complemented by other methods—both holistic and atomistic—that focus on other mechanisms, such as electrostatic isopotentials [16]. Together, these tools might assemble explanations for mechanisms that achieve specific binding.

HBcompare classifies patterns of hydrogen bonds using graph convolutional networks (GCNs), which make use of the symmetrically normalized graph Laplacian to compute vertex embeddings and to evaluate vertex similarity [19]. Recent works [20,21] have shown that GCNs are useful for automating feature learning from graph-structured data compared to traditional methods, such as convolutional neural networks (CNN). HBcompare adapts existing GCN approaches by constructing a molecular graph for each protein to aggregate neighborhood information. As a result, HBcompare performs accurate graph classification and avoids sensitivity to the input order of graph vertices, which can be a challenge for existing methods.

In this paper, we evaluated the effectiveness of HBcompare for classifying protein binding preferences on several protein superfamilies. Each superfamily was selected because it contained well defined subfamilies with different binding preferences, where differences in specificity hinge on differences in hydrogen bonding patterns. These superfamilies include groups of subfamilies from the tRNA-synthetases, the alpha-amylases, and the serine proteases. Our computational results explore how accurately HBcompare performed classifications consistent with experimentally established binding preferences. We also examine how HBcompare would perform in a more holistic setting, integrated with atomic coordinates, and compare its performance to existing methods on the same kinds of features. These results point to the importance of considering the distinct applications of both holistic and atomistic techniques.

## 2. Methods

### 2.1. Constructing Molecular Graphs with HBondFinder

HBcompare represents hydrogen bond topologies using molecular graphs. We define a molecular graph as an undirected graph G=(V,E,A). The nodes or vertices $V={\left\{v_{i}\right\}}_{i=1}^{N}$ are atoms that are hydrogen bond donors and/or acceptors. The edges *E* are hydrogen bonds, identified between one donor and one acceptor atom. Since donors and acceptors may be positioned to participate in one of several possible hydrogen bonds, the resulting graph may be more than a collection of disconnected donor-acceptor pairs. Finally, a weighted adjacency matrix A describes the weights $A_{ij}$ of edges between nodes *i* and *j*.

To generate molecular graphs from protein structures, we developed HBondFinder, which uses geometric criteria to determine the set of all possible hydrogen bonds. Beginning with a standard chain from the Protein Data Bank [22], we prepare the data by first removing all ligands, ions, hydrogens and water molecules. Hydrogens specifically are removed because their positions are not always solved in an experimental crystal structure, leaving some amino acids with incomplete protonation. Thus, for uniformity, we model the positions of all hydrogens using the reduce tool from MolProbity [23], assuming biological pH. We then use the element of each atom, its position within an amino acid and residue names, which define the type of amino acid, to identify all atoms that are hydrogen bond donors, donor hydrogens, hydrogen bond acceptors, and acceptor antecedants. These four atoms appear in pairs on each end of the hydrogen bond. The nodes of the molecular graph are defined by any atom that is a donor, acceptor, or both.

HBondFinder defines the edges of the graph by finding all donor-acceptor pairs that satisfy our hydrogen bond criteria, which are inspired by the HBPlus program [24]. This process is accelerated with a lattice-based geometric data structure [25] that allows us to rapidly search for all atoms of a specific identity that are within a radius of a given point. This search allows us to find all combinations of the four critical atoms of a hydrogen bond: "D", the hydrogen bond donor, "H", the donor hydrogen, "A", the acceptor, and "AA", the acceptor antecedent. From these combinations, we enforce our criteria: First, the D-A distance must be within 3.9 Å, and the H-A distance must be within 2.5 Å. In addition, the angles D-H-A, H-A-AA, D-A-AA, where the middle member is the node of the angle, must all exceed 90 degrees. If these four atoms satisfy the constraints, then a hydrogen bond could exist and we add an edge to the graph, and a weight of 1.0 to the adjacency matrix, between donor and acceptor. All weights on the adjacency matrix are otherwise zero. We refer to graphs with these binary weights as *coordinate-free molecular graphs*.

To compare the predictive value of coordinate-free molecular graphs to a maximally similar representation that incorporates atomic coordinates, we also created a second kind of molecular graph called a *coordinate-based molecular graph*. These graphs are identical except that the edges recorded in the adjacency matrix, between donors and acceptors that can form a hydrogen bond, are weighted by the Euclidean distance in angstroms.

### 2.2. HBcompare

**Overview.** We hypothesize that molecular graphs with similar topology and class labels will describe proteins with similar binding preferences. These proteins are expected to exhibit different numbers of atoms, different amino acids, different numbers of hydrogen bond donors and acceptors, and also some variation in edge topology. The classification task performed by HBcompare begins with a set of molecular graphs $\left\{G_{1},\cdots ,G_{M}\right\}$, each assigned a subfamily class label ${\left\{y_{i}\right\}}_{i=1}^{M}$. HBcompare performs whole-graph analysis on an input graph $G_{i}$ to learn an embedding $e_{G_{i}}$ and predict its subfamily label $y_{i}$ (Figure 1).

Consider the general multi-layer GCN model with the following propagation rule for graph-structured data [19]:(1)$$X^{\left(l\right)}=\sigma \left(\hat{A}X^{\left(l-1\right)}W^{\left(l\right)}\right),$$
where $\hat{A}\in R^{N\times N}$ is the normalized adjacency matrix of the graph *G* with added self-connections, i.e., $\hat{A}=D^{-\frac{1}{2}}\left(A+I_{N}\right)D^{-\frac{1}{2}}$, D is the degree matrix, $W^{\left(l\right)}\in R^{D^{\left(l-1\right)}\times D^{\left(l\right)}}$ is the layer-specific weight matrix with trainable parameters, and σ(·) is a nonlinear activation function. $X^{\left(l-1\right)}\in R^{N\times D^{\left(l-1\right)}}$ is the input of the *l*-th layer, and $X^{\left(l\right)}\in R^{N\times D^{\left(l\right)}}$ is the output of the *l*-th layer. Naturally, $X^{\left(0\right)}$ is the initial node feature matrix.

In the following, we show how the propagation rule of GCN in Equation (Equation 1) can be extended to multiplex models, thereby enabling HBcompare to learn graph representations across multiple graphs with different orders and sizes of nodes.

**Node feature construction.** Unfortunately, the initial node features are not available. To solve this issue, we notice that every node of a molecular graph is labelled a hydrogen bond donor, acceptor, or both, we adopted the one-hot encoding strategy on node labels [26] to construct the input node feature matrix $X^{\left(0\right)}\in R^{N\times 3}$.

**Multi-GCN model.** After the initial node representations are obtained, each molecular graph can be represented by $G=(V,E,A,X^{\left(0\right)})$. To explain how the multi-GCN model works, we first analyze the propagation Equation (Equation 1) and factorize it into feature aggregation (FA) and feature transformation (FT) following [27].

*Feature aggregation.* To learn the node representation $X^{\left(l\right)}$ of the *l*-th layer, in the first step GCN follows the neighborhood aggregation strategy to smooth nodes’ representations over a graph by
(2)$${\hat{X}}^{\left(l\right)}=\hat{A}X^{\left(l-1\right)},$$

This means that the role of $\hat{A}$ in GCN is to aggregate the neighborhood information of a node for updating its embedding. This design of GCN is suitable for hydrogen bond data analysis. First, the learning process and the ultimate classification of graphs with similar topologies is performed independent of the order in which the nodes are described. Second, the GCN approach is unaffected by graphs with sparse edges, where classification is more difficult. Finally, noise in hydrogen positions, which may affect whether a hydrogen bond is considered to exist near its length and angle limits, is also unlikely to affect classification.

*Feature transformation.* After FA, in the second step GCN conducts FT in the *l*-th layer, which consists of linear and nonlinear transformations:(3)$$X^{\left(l\right)}=\sigma \left({\hat{X}}^{\left(l\right)}W^{\left(l\right)}\right)$$

The weight matrix $W^{\left(l\right)}$ can adjust the output features, which is equivalent to feature selection and combination. Intuitively, if the same weight matrix is used for different graphs, then we can project them into the common feature space with the same dimension to perform group analysis.

Based on the above analysis, we generalize the propagation rule in Equation (Equation 1) to the following form for multi-graph embedding.
(4)$$X_{i}^{\left(l\right)}=\sigma \left({\hat{A}}_{i}X_{i}^{\left(l-1\right)}W^{\left(l\right)}\right),\;\forall i\in \left\{1,2,\cdots ,M\right\}$$
where ${\hat{A}}_{i}$ is the normalized adjacency matrix of the *i*-th graph $G_{i}$, $X_{i}^{\left(l-1\right)}$ and $X_{i}^{\left(l\right)}$ are its corresponding input and output embeddings of nodes in the *l*-th layer, and $W^{\left(l\right)}$ is the trainable weight matrix shared by all graphs.

To obtain the vector representation $e_{G_{i}}$ of the entire graph $G_{i}$, a general and straightforward practice [28,29] is to aggregate the embedded node features of the last GCN layer. However, the extracted information from each layer could also be useful to supplement the graph structure—especially for the molecular graphs that are sparse and the initial information of nodes is not rich. Thus, we adopt the concatenation strategy [30] to exploit features from all layers at multiple scales to contribute to the characterization of the graph, and let the classifier decide which of the features are useful. More specifically, we concatenate the node features $X_{i}^{\left(l\right)}$ from all layers to get the final node representation matrix
(5)$$X_{i}^{all}=\left[X_{i}^{\left(1\right)},X_{i}^{\left(2\right)},\cdots ,X_{i}^{\left(L\right)}\right],$$
where $X_{i}^{all}\in R^{N_{i}\times \sum_{l=1}^{L}D^{\left(l\right)}}$, with each row corresponding to a node and each column corresponding to a feature, and $N_{i}=\left|V_{i}\right|$ is the number of nodes for the *i*-th graph $G_{i}$.

**Whole-graph training.** Based on the node representations, we are able to design different task-specific loss functions to train the overall multi-GCN model in the same way as of training GCN. Since for the protein family identification problem, we have access to all nodes from the entire datasets and the node labels are available, we can adopt the learning method in [20,31] to make full use of the node-level information and also capture the substructures within each graph to improve classification accuracies. Specifically, given the node label set Y for all nodes, the training process for multi-GCN is then formulated as:(6)$$\min_{W^{\left(1\right)},\cdots ,W^{\left(L\right)},\Theta }Loss\left(\left\{X_{1}^{all},\cdots ,X_{M}^{all}\right\},\Theta ,Y\right),$$
where $\Theta \in R^{\sum_{l=1}^{L}D^{\left(l\right)}\times C}$ is the linear classification matrix, *C* is the number of classes in the classification problem, and Loss(·) is the cross-entropy loss function for multi-class classification.

**Graph embedding and classification.** There are several ways to get the graph-level outputs using node features, such as concatenation, mean pooling, and max pooling operators [29]. In our task, graphs are not aligned across different subjects and each graph may have an arbitrary number of nodes. Thus, the average pooling technique is used [30] here to obtain the embedding $e_{G_{i}}$ of the entire graph $G_{i}$, which allows us to eliminate the dependence on the node order and size. Mathematically, for each graph $G_{i}$, we can formalize the mean pooling of node features as
(7)$$e_{G_{i}}=\frac{1}{N_{i}}\sum_{v\in V_{i}}\left[x_{i_{v}}^{\left(1\right)},x_{i_{v}}^{\left(2\right)},\cdots ,x_{i_{v}}^{\left(L\right)}\right],\;\forall i\in \left\{1,2,\cdots ,M\right\}$$

Finally, we apply the logistic regression (LR) classifier based on the above whole-graph embedding vectors ${\left\{e_{G_{i}}\right\}}_{i=1}^{M}$ and associated protein subfamily class labels ${\left\{y_{i}\right\}}_{i=1}^{M}$ as input for prediction.

### 2.3. Datasets Used in This Study

To evaluate HBcompare as a classifier, we constructed datasets based on protein superfamilies with three criteria. First, we selected superfamilies that contained subfamilies with distinct ligand binding preferences. Second, we selected only superfamilies and subfamilies where differences in binding preferences are experimentally established to rely on variations in hydrogen bonding patterns. Finally, proteins in each superfamily were selected with the same overall fold.

These criteria enable our datasets to test the overall hypothesis. The first two criteria are required for evaluating HBcompare as a classifier of hydrogen bonding topologies. The third ensures that the classification task is not trivial, because subfamilies with different folds have very different hydrogen bond topologies that can be easily distinguished. The general properties of the constructed protein datasets are summarized in Table 1 and details are described as below.

**Primary protein datasets.** Our criteria identified the glycosidases, the serine proteases, the aminoacyl-tRNA synthetases, and several subfamilies of each (Table 1). We used the Enzyme Commission Classification index [32] of each subfamily to identify the protein data bank (PDB) [33] structure of every constituent protein. To avoid the overrepresentation of well studied proteins with many available structures, we removed one member of any pair of proteins with greater than 95% sequence identity. We also removed any structures labeled as mutants to avoid misclassifying proteins with deactivating mutations (Table 2). After this filtration, molecular graphs were generated on the remaining structures using the method in Section 2.1.

There are 303 structures across all primary datasets. 298 structures were derived from X-ray crystallography, and five were produced by nuclear magnetic resonance spectroscopy. Xray structure resolutions ranged from 0.81 Å to 3.5 Å, with an average of 2.05 Å, a median of 2.0 Å, and a standard deviation 0.443 Å. 291 out of 303 structures have resolution less than or equal to 3.0 A, and 261 out of 303 structures have resolution less than or equal to 2.5 A. The number of proteins observed in each subfamily of each dataset was generally similar, requiring no additional treatment to to balance the datasets.

In Primary-1 (P1), the glycosidase superfamily proteins conserve an alpha/beta barrel fold where they hydrolyze the glycosidic bonds of polysaccharide chains. The alpha and beta amylase subfamilies hydrolyze the intermediate and the terminal bonds, respectively, of these chains, and recognize them in part through differences in hydrogen bonding [34,35].

In Primary-2 (P2), the PA clan of the serine protease superfamily exhibit a chymotrypsin-like fold and catalyze the cleavage of peptide bonds. They share a catalytic triad at the center of an extensive hydrogen bonding network that also plays a crucial role in stabilizing substrate backbones for efficient substrate hydrolysis [36].

In Primary-3 (P3), the aminoacyl-tRNA synthetases catalyze the attachment of a transfer RNA and an amino acid in preparation for protein translation. The seryl- and threonyl-tRNA Synthetase share an anti-parallel beta-sheet fold [37] but coordinate their amino acid substrates through different patterns in hydrogen bonding [38,39].

**Auxiliary datasets.** We also developed two variations on our original datasets to evaluate the performance of HBcompare. Noting that the serine protease dataset has five subfamilies, we developed a two-subfamily variation, using only the chymotrypsin and trypsin subfamilies. This variation allowed us to evaluate how HBcompare performed on a classification problems with different numbers of categories. We created a second dataset to evaluate the scenario where some subfamilies have different folds, and thus radically different hydrogen bond topologies. We combined the glycosidases and the serine proteases into a single artificial superfamily. Using two subfamilies of each of the joined superfamilies, we assess if the substantial differences between the superfamilies obscure the subtler differences between subfamilies.

### 2.4. Comparison with Existing Methods

Directly comparing HBcompare against existing methods is difficult, because HBcompare uses only the topology of hydrogen bonds while existing methods for comparing protein structures generally require atomic coordinates and other data. For this reason, we performed two separate comparisons. First, to demonstrate the fitness of HBcompare as a tool for coordinate-free graph classification, we compare the performance of HBcompare against several modern graph classification techniques that also use only graph topology. Second, to understand how classification by hydrogen bond topology performs relative to classification by atomic coordinates, we modified all methods, including HBcompare, to incorporate coordinate-based molecular graphs (see Section 2.1).

Our first comparison study includes a convolutional neural network (CNN), a graph kernel-based comparison method (GK), and principal component analysis based methods (PCA, 2DPCA, and PCA-NF). These methods use hydrogen bond topology alone via an analysis of node adjacency matrices, but they have never been applied for the coordinate-free comparison of hydrogen bond topologies. As such, they require modifications for direct comparison. The need for small modifications demonstrates, qualitatively, a degree of unsuitability for the problem of topological comparison relative to HBcompare, which does not require such modification.

First, CNN, PCA and 2DPCA are sensitive to variations in input order, while GCNs are not. To minimize this sensitivity, dataset proteins were structurally aligned to an arbitrarily selected pivot structure to produce a 1-to-1 mapping between most amino acids, ensuring that all proteins could be indexed in the same order. Structural alignments were performed with ska [1], which is designed for identifying distantly related proteins with subtle similarities in their folds. In this application, where we are considering datasets of closely related proteins with nearly identical folds, ska easily generated 1-to-1 mappings appropriate for our comparison.

Second, CNN, PCA and 2DPCA also require input data to have the same number of nodes, because the features they consider cannot have varying dimensionality. To resolve this issue, we trimmed all molecular graphs to contain exactly 600 nodes, a quantity chosen because the largest connected component of all graphs in our dataset would not be altered. This trimming was possible without disrupting the topology of the graph because all structures contain a large number of donors and acceptors that are uninvolved in a hydrogen bond. In the molecular graph they are singleton nodes, and they contribute no distinguishing information to the topological character of the graph overall. By removing some of these nodes as necessary, we were able to trim larger graphs to exactly 600 nodes. Graphs that had fewer than 600 nodes, such as those in P2, had singleton nodes added to arrive at exactly 600 nodes.

Our second comparison study adds the protein structure comparison algorithm Ska and the sequence comparison algorithm Clustalw [40]. These classic methods benchmark the performance of HBcompare against existing comparison techniques in structural bioinformatics. GK, CNN, PCA and 2DPCA remain, but they are provided coordinate-based rather than coordinate-free molecular graphs as input.

The CNN model [41] utilizes shared weights for common feature extraction, and also local reception fields to take advantage of the local structure of input data. In our case, we trained an end-to-end CNN model with fully connected network (FCN) classifier that takes adjacency matrices A as input and outputs the corresponding graph classes.

The GK method [42] applies the Weisfeiler Lehman (WL) kernel to calculate similarities between graphs [43,44]. Each vertex is labelled with its original vertex label and the label of its neighbors, resulting in a representation of graphical neighborhoods of each vertex. The WL kernel goes through *n* iterations until WL kernels are unchanged for successive iterations. This kernel is then fed into a support vector machine (SVM) to measure the graph classification performance.

The PCA method [45] for comparing graphs learns a common projection matrix via singular value decomposition (SVD) by vectorizing the submatrices to perform feature extraction. Similar to our HBCompare model, the extracted graph feature vectors are passed to the LR classifier. Furthermore, to investigate the effectiveness of using one-hot encoding labels as the node feature input for GCN, we also concatenate the features extracted by PCA and the GCN node features. This variation, PCA-NF, adds the donor/acceptor status of each graph node to the topology being classified.

The 2DPCA method [46] avoids vectorization of input submatrices by learning pairwise projection matrices for feature extraction and dimensionality reduction. The extracted feature matrices are then vectorized and fed to the LR classifier for prediction.

The ska [1] algorithm finds corresponding secondary structure elements between two proteins to build detailed correspondences between backbone atom coordinates, which are required. The atomic correspondences are used to compute least root mean square difference (RMSD) between backbone atoms. As a measure of geometric similarity, RMSD is lower between proteins that are more similar. Using ska, we generated an all-vs-all matrix of RMSD distances between all proteins of each dataset. Viewed as a set of column vectors, this matrix is decomposed into training and test sets and the training sets are used to train an LR classifier via five fold validation, similar to [47]. Finally, the test set is passed to the classifier to form predictions.

Clustalw [40] is the classic sequence-based comparison algorithm that measures similarity between the sequences of amino acids that define two proteins. It applies dynamic programming to build correspondences between amino acid sequences and then measure the percentage of sequence identity. Higher percentages are generated by protein pairs with similar sequences of amino acids, and lower percentages indicate proteins that are more different. These percentages are subtracted from 100 so that smaller values indicate more similar proteins, and then used to populate an all-vs-all matrix that is treated in the same way as the RMSDs are for ska.

### 2.5. Implementation Details

All models were implemented in Python 3.6 with Tensorflow 1.15 for the deep learning backend. The validation of our method was performed by randomly and uniformly splitting each dataset and each subclass by a 4:1 ratio. The split results in a larger training set (80% of the data) and a smaller test set (20% of the data). Since the subclasses were split uniformly, the approximate balance of the subclasses in each dataset was preserved in each split. The performance of all classifiers reported in Tables 3 and 4 is an average and a standard deviation of accuracy, f1-score, and AUC-ROC computed from 10 such random splits. We evaluate predictions as correct if the prediction agrees with the class label and incorrect if the prediction does not agree with the label. We report accuracy (acc) as the ratio of correct predictions to total predictions, $\frac{Correct}{Correct+Incorrect}$.

We performed parameter tuning on all methods using 5-fold validation on the training set. Since this training set is held separate from the testing set, no data leakage influences the classifier performance reported. Training was performed for 50 epochs per fold, and parameters associated with the highest accuracy fold were used for evaluation on the corresponding test set. We used the Adam optimizer [48] and selected the learning rate $l_{r}$ from $\left\{5e^{-4},1e^{-4},5e^{-3},1e^{-3}\right\}$.

For the design of HBcompare we considered between 1 and 6 GCN layers, and batch sizes in the range {1,2,4,8,16}. To build the CNN model, we varied the number of filters in the set {6,12,18,24,30}, and the number of strides in the set {1,2,4,8,16}. The total number of parameters in the network was 384. The number of layers, epochs, the batch size, and learning rate are selected for the CNN model in the same manner as HBcompare. For the other compared methods, we also carefully tuned their parameters and use the same data splits and the same 5-fold cross-validation scheme. All experiments were performed on a 8-core machine with 32 GB RAM.

## 3. Results

During training, we observed converging improvements in accuracy relative to training time and number of epochs. These observations are illustrated for all datasets in Appendix A, Figure A1 and Figure A2. By dividing the data sets into non-overlapping training and testing sets, we found that classification accuracy of HBcompare for training and testing quickly converged towards a stable accuracy performance and remains at this performance level regardless of added epochs past the saturation point. This is shown in Appendix A, Figure A3. Collectively, these observations suggest that overtraining is not a major concern for the accuracy of HBcompare on our datasets.

Overall, using only hydrogen bond topology, HBcompare had a high degree of classification accuracy. The classification accuracy of HBcompare averaged from 85.0% to 92.3% on all folds of all primary datasets (Table 3, right column, top three rows). The standard deviation in accuracy across all folds ranged from 4.8% to 7.7%. The F1 score averaged between 84.8% and 92.2%, and the area under the ROC curve (AUC-ROC) averaged between 90.6% and 92.3%.

In comparison to existing coordinate-free methods, HBcompare was 11.38% more accurate, had 12.17% greater F1 score, and had 9.92% higher AUC, on average, than the second best method, PCA-NF, across all data sets. Standard deviations in HBcompare accuracy, F1 score and AUC were also generally the same or lower than existing methods. Overall, HBcompare had the best classification performance of all methods on all primary datasets (Table 3, top three rows).

Auxiliary-1 simplified the multi-class classification problem by removing three of the five subfamilies in Primary-2. As a result, on Auxiliary-1, all comparison methods were significantly more accurate, with PCA-NF outperforming HBcompare slightly (93.8% vs. 91.8%). The fact that HBcompare significantly outperforms other methods on the five categories of Primary-2 suggest that it is more robust to the multi-class classification problem.

On Auxiliary-2, which combined two subfamilies from each of Primary-1 and Auxiliary-1, HBcompare outperformed other methods by at least 6.8%. In this case, where some subfamilies are far more similar than others, HBcompare did not lose discriminating power, performing only slightly worse than it did on Primary-1 and on Auxiliary-1 despite two additional categories.

Since HBcompare operates with only hydrogen bond topology, we also asked how HBcompare and other graph-based methods would perform if atomic coordinates were included (Table 4). Again, on all primary datasets, HBcompare outperformed existing methods, with accuracy averaging from 2.1% to 14.9% above existing methods. Unsurprisingly, since these comparisons used representations of both hydrogen bond topology and also atomic coordinates, GK, PCA, 2DPCA, CNN, and HBcompare all performed the same or better than their coordinate-free counterparts. Classifications using only sequence identity or structure similarity underperformed.

On Auxiliary-1, the addition of atomic coordinates into the graph representation resulted in slightly superior classification accuracy for HBcompare (93.8%) relative to PCA-NF (91.3%). As in the coordinate-free scenario, GK, PCA, 2DPCA and CNN all performed similarly. On Auxiliary-2, HBcompare was again more accurate (88.4%).

### 3.1. Hyperparameter Analysis

In training HBcompare, we considered a range of batch sizes and GCN layers, both of which can influence classifier performance. Adding more GCN layers expands the graph neighbourhood within which the node features are averaged [49]. These findings are plotted in Figure 2. We observed that accuracy was maximized with batch size 4 and with 3 GCN layers, using these parameters in HBcompare.

### 3.2. Feature Concatenation

In our HBcompare model, we concatenate the output of all GCN layers to obtain the final feature representation (Figure 1). To evaluate the effectiveness of this concatenation strategy, we compare the implementation of HBcompare with and without feature concatenation in Table 5 using only hydrogen bond topology. We observed that HBcompare can benefit from the concatenation strategy, which helps to aggregate more information when the input node feature size is small.

## 4. Discussion

We have presented HBcompare, a GCN-based algorithm for classifying protein structures based exclusively on hydrogen bonding topology. Once trained on a group of closely related subfamilies that perform the same function on different preferred ligands, HBcompare addresses the problem where a novel protein structure or model is to be classified into one of the subfamilies. HBcompare should be retrained to make classifications into different subfamilies.

Since it only examines hydrogen bond topology, accurate classifications implicate the importance of hydrogen bonds in achieving the binding preferences of the predicted subfamily. This novel capability contrasts from holistic representations, which do not implicate specific mechanisms.

To evaluate HBcompare, we performed specificity classification experiments on protein superfamilies that achieve distinct binding preferences based on differences in hydrogen bonding. On nonredundant subsets of the glucosidases, serine proteases, and tRNA synthetase superfamilies, the average accuracy of HBcompare was 92.3%, 85.0% and 91.3%. As a tool for classifying hydrogen bond topologies, HBcompare is a capable classifier. When we adapted several modern techniques to the topology-only classification problem, we observed that HBcompare was more accurate in all but one case, where PCA with node features outperformed HBcompare 93.8% versus 91.8%. This classification performance was well within the variations observed in different training folds, indicating comparable performance between PCA-NF and HBcompare, rather than a superior performance of one over the other. Furthermore, it is important to note that CNN, GK, PCA, PCA-NF and 2DPCA all require a structural alignment to produce a 1-to-1 mapping between most amino acids, ensuring that all proteins could be indexed in the same order. CNN, PCA, PCA-NF and 2DPCA also require input graphs to have the same number of nodes. Our comparison included a preprocessing step that maximizes their comparability in this study, but in truly experimental settings, accurate preprocessing could not be guaranteed, further limiting the applicability of these alternative methods. The same challenges do not apply to HBcompare, which is unaffected by input order or graph size, making it more applicable in experimental settings and often more accurate than existing methods.

We also compared HBcompare to conventional coordinate-based approaches. In comparison to ska, a coordinate-based method that does not use hydrogen bonding topology (Table 4), coordinate-free HBcompare (Table 3) was an average of 20.6% more accurate on all datasets. These findings demonstrate that hydrogen bond topology contributes information that is complementary to conventional structural approaches.

Finally, we modified HBcompare to consider both atomic coordinates and also hydrogen bond topology. In a comparison to the same methods above, each modified to incorporate both data types, HBcompare was 2.1% to 14.9% more accurate on average (Table 4). This result demonstrates that combining hydrogen bond topology and atomic coordinates enhances subfamily classification at the cost of being able to implicate hydrogen bonds as a mechanism.

As a first step in the atomistic analysis of hydrogen bond topology, HBcompare has considerable potential for novel applications. Where specificity mechanisms are unknown, HBcompare can detect when hydrogen bonding distinguishes between isoforms with different binding preferences without influences from other structural properties. This capability can focus experimental scrutiny on hydrogen bonding when it correlates with specificity. Combined with structural models, HBcompare could be applied to identify mutations that change bond topology to resemble proteins with different binding preferences. Together with other sources of information, HBcompare could thus support efforts in protein engineering and in annotating binding specificity mechanisms.

## Figures and Tables

**Figure 1 biomolecules-12-01589-f001:**
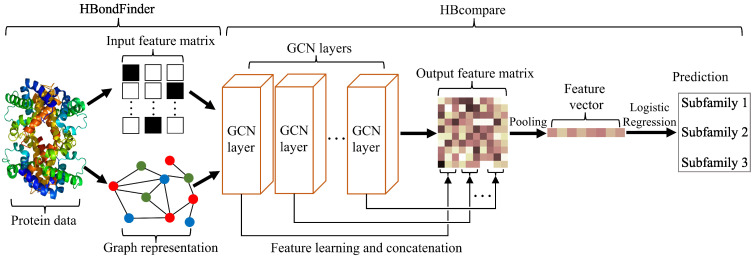
The HBcompare model. As input, HBondFinder takes protein structures and constructs the feature matrix and graph representation. Next, these data are analyzed using GCN layers and their results are concatenated to generate the output feature matrix, which is vectorized via graph pooling and fed to a logistic regression (LR) classifier.

**Figure 2 biomolecules-12-01589-f002:**
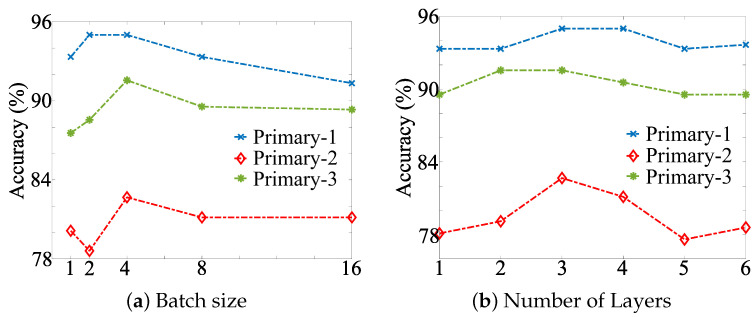
Influence of the number of layers (**a**), and of the batch size (**b**) on the classification accuracy of HBcompare. Accuracy is shown on all three primary datasets (blue, red and green lines), and was highest for batch size 4 and for 3 GCN layers.

**Table 1 biomolecules-12-01589-t001:** Primary and Auxiliary data sets used in this study.

Dataset	Superfamily E.C. Class	Pivot Structure	Subfamily	E.C. Class	Number of Structures
Primary-1 (P1)	Glycosidases 3.2.1.*	1aqm	Alpha Amylase Beta Amylase	3.2.1.1 3.2.1.2	30 30
Primary-2 (P2)	Serine Proteases 3.4.21.*	1ghz	Chymotrypsin	3.4.21.1	40
			Trypsin	3.4.21.4	40
			Elastase	3.4.21.36	40
			Thrombin	3.4.21.5	37
			Coagulation	3.4.21.6	39
			factor Xa		
Primary-3 (P3)	Aminoacyl-tRNA Synthetases 6.1.1.*	6rlt	Ser-tRNA Synthetase Thr-tRNA Synthetase	6.1.1.11 6.1.1.3	24 23
Auxiliary-1 (A1)	Serine Proteases (subset)	1ggd	Chymotrypsin Trypsin	3.4.21.1 3.4.21.4	40 40
Auxiliary-2 (A2)	Glycosidases, Serine Proteases (A1 + P1)	2xfy	Alpha Amylase Beta Amylase Chymotrypsin Trypsin	3.2.1.1 3.2.1.2 3.4.21.1 3.4.21.4	40 40 30 30

**Table 2 biomolecules-12-01589-t002:** Average properties of proteins in all datasets.

Dataset	# Proteins	# Subfamilies	Avg. # Nodes	Avg. # Edges
P1	60	2	826	578
P2	196	5	402	241
P3	47	2	901	573
A1	80	2	372	201
A2	140	4	568	363

**Table 3 biomolecules-12-01589-t003:** Average classification accuracy and F1 score (avg ± std) of compared methods using only hydrogen bond topology, across all cross-validation folds. The set(#) column indicates the dataset and the number of subfamilies it contains. The *stat* column indicates rows with either classifier accuracy or F1 score. The highest value in each row is bolded.

set(#)	stat	GK	PCA	PCA-NF	2DPCA	CNN	HBCompare
	Acc	68.8 ± 1.1	76.7 ± 15.3	81.7 ± 12.2	73.3 ± 8.2	85.0 ± 15.3	**92.3 ± 7.0**
P1(2)	F1	69.3 ± 1.1	74.7 ± 17.6	79.8 ± 15.6	73.1 ± 8.1	83.1 ± 18.5	**92.2 ± 6.8**
	AUC-ROC	68.8 ± 1.1	76.7 ± 15.3	81.7 ± 12.2	73.3 ± 8.2	85.0 ± 15.3	**92.3 ± 6.7**
	Acc	38.1 ± 1.5	63.3 ± 4.6	67.8 ± 4.0	65.3 ± 6.9	68.6 ± 3.1	**85.0 ± 4.8**
P2(5)	F1	41.8 ± 1.4	62.0 ± 5.0	67.2 ± 3.9	65.4 ± 7.0	68.6 ± 3.4	**84.8 ± 5.0**
	AUC-ROC	27.0 ± 1.1	76.8 ± 2.8	79.7 ± 2.4	69.1 ± 4.3	80.2 ± 2.0	**90.6 ± 3.1**
	Acc	58.7 ± 3.4	61.6 ± 10.7	68.0 ± 9.6	59.1±12.0	68.0 ± 9.6	**91.3 ± 7.7**
P3(2)	F1	60.2 ± 3.7	60.9 ± 10.8	66.3 ± 10.7	58.4 ± 12.7	67.3 ± 9.2	**91.2 ± 8.5**
	AUC-ROC	58.1 ± 3.4	62.0 ± 10.2	67.0 ± 9.9	60.0 ± 11.3	68.0 ± 8.6	**91.5 ± 8.2**
	Acc	76.6 ± 1.2	90.0 ± 5.0	**93.8 ± 4.0**	90.0 ± 3.1	88.1 ± 4.4	91.8 ± 5.5
A1(2)	F1	77.8 ± 1.5	89.9 ± 5.0	**93.7 ± 4.0**	89.9±3.1	88.0 ± 4.4	91.7 ± 5.6
	AUC-ROC	76.6 ± 1.2	90.0 ± 5.0	**93.8 ± 4.0**	90.0 ± 3.1	88.1 ± 4.4	91.8 ± 5.5
	Acc	52.2 ± 0.9	75.7 ± 4.7	80.0 ± 2.9	72.9 ± 5.3	73.6 ± 3.6	**86.8 ± 5.4**
A2(4)	F1	50.2 ± 2.0	74.3 ± 5.2	79.3 ± 3.1	70.1 ± 5.1	73.3 ± 3.6	**86.3 ± 6.7**
	AUC-ROC	54.6 ± 1.0	83.4 ± 3.3	86.3 ± 2.1	81.3 ± 3.0	82.4 ± 1.7	**90.9 ± 4.3**

**Table 4 biomolecules-12-01589-t004:** Average classification accuracy and F1 score (avg ± std) using both hydrogen bond topology and coordinate information, across all folds. The *set* column indicates the dataset. The *stat* column indicates rows with either classifier accuracy or F1 score. The highest value in each row is bolded.

Set	stat	Clustalw	Ska	GK	PCA	PCA-NF	2DPCA	CNN	HBcompare
P1	Acc	75.0 ± 17.7	63.3 ± 4.6	68.3 ± 1.2	83.3 ± 12.9	86.7 ± 11.3	83.3 ± 9.1	88.3 ± 8.5	**92.3 ± 7.2**
	F1	79.5 ± 11.7	66.9 ± 5.3	68.8 ± 1.2	81.5 ± 16.3	85.8 ± 12.8	82.4 ± 10.6	87.8 ± 9.6	**92.8 ± 7.1**
P2	Acc	80.0 ± 8.1	68.8 ± 7.7	37.3 ± 1.1	70.4 ± 3.8	72.4 ± 3.9	74.5 ± 5.9	70.4 ± 3.0	**83.6 ± 6.3**
	F1	80.8 ± 7.1	70.1 ± 8.1	41.7 ± 1.8	69.5 ± 5.5	72.0 ± 4.7	74.8 ± 5.6	70.8 ± 3.1	**83.9 ± 6.5**
P3	Acc	60.8 ± 3.4	63.3 ± 2.6	58.7 ± 2.4	74.7 ± 12.8	76.7 ± 12.6	74.7 ± 12.8	76.8 ± 8.9	**90.6 ± 6.8**
	F1	69.2 ± 2.3	73.4 ± 7.7	60.3 ± 2.8	73.8 ± 13.1	76.2 ± 12.7	73.8 ± 13.1	76.4 ± 8.7	**90.5 ± 6.9**
A1	Acc	60.7 ± 2.5	50.0 ± 8.1	76.3 ± 1.3	91.3 ± 6.4	91.3 ± 6.4	92.5 ± 6.1	91.9 ± 5.6	**93.8 ± 4.4**
	F1	67.6 ± 3.8	53.9 ± 6.8	77.3 ± 1.5	91.2 ± 6.4	91.2 ± 6.4	92.5 ± 6.1	91.8 ± 5.6	**93.9 ± 4.4**
A2	Acc	86.6 ± 14.5	81.1 ± 8.0	52.1 ± 0.5	75.0 ± 6.4	77.9 ± 5.2	80.7 ± 5.3	80.4 ± 6.4	**88.4 ± 6.4**
	F1	87.6 ± 13.3	82.3 ± 6.8	49.5 ± 1.3	73.7 ± 6.5	76.7 ± 5.4	80.0 ± 5.4	80.1 ± 6.7	**88.2 ± 6.5**

**Table 5 biomolecules-12-01589-t005:** Average classification accuracy of HBcompare model with and without concatenation strategy using only hydrogen bond topology across all folds. The more accurate method is bolded.

Method (Acc)	Primary-1	Primary-2	Primary-3	Auxillary-1	Auxillary-2
Hbcompare with concatenation	**91.3**	**80.6**	**89.6**	90.8	**87.1**
Hbcompare without concatenation	88.6	78.2	89.1	**91.8**	86.4

## Data Availability

HBcompare and HbondFinder have been made open source. The five datasets used in this study are available here. These links were accessed on 28 Oct 2022.

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
