# Peer review of "HBcompare: Classifying Ligand Binding Preferences with Hydrogen Bond Topology"

_biomolecules, 2022, doi:10.3390/biom12111589_

Round 1
Reviewer 1 Report
This paper introduces a tool, hbCompare, that uses hydrogen bonding as determined from experimental structures to classify binding preferences.
Section 2.4 discusses locating the densest subgraph. If I understand "densest", then the max-clique NP problem could be reduced to the technique proposed in the paper. Some high-level details of how this is computed should be mentioned.
Section 2.5 (implementation details) -- While some details are given. The paper should cite that tensorflow was used (although can be determined by looking at the github code). The paper does not indicate how many variables/parameters were used in the network, which is not obvious given the GCN structure. What were the wall clock times for the number of epochs used for each dataset studied? On a positive note, figure 2 does a nice job of showcasing how two hyperparameters influence accuracy across a few datasets.
The datasets that are assembled and summarized in Table 2 are all experimentally determined high-resolution structures? What is the min-max resolution? Are they all x-ray or are some NMR determined? For those that are NMR determined, what, if any, variance existed in the hydrogen locations?
How sensitive is the classifier to the locations of the hydrogen atoms? If noise is introduced into the dataset (in the coordinates), how does it impact the results? The coordinate-free and coordinate approaches could both be impacted by this noise. This type of analysis seems very relevant since a single structure in some cases may adequately characterize the conformational selection approach to binding.
Small grammar correction: Section 2.3 on auxiliary datasets, "has a five subfamilies" should be "has five subfamilies".
Author Response
(please see attachment)

Reviewer 2 Report
The manuscript “HBcompare: Classifying ligand binding preferences with hydrogen bond topology” addresses the fundamental challenge of protein function annotation from structure. The authors describe a new classifier, based solely on hydrogen bond topology, of protein subfamilies sharing a similar enzyme reaction mechanism but with different ligands. To this aim, the authors used graph convolutional networks to classify several protein subfamilies belonging to five different protein families. They compared the performance of the new method with others and found comparable or superior classification accuracy.
Comments:
- The performance of the classifiers was assessed with the accuracy and F1 scores, but it would be also useful to provide the frequently used area under the receiver operating characteristic curve (AUC ROC).
- In table 2 I would suggest the authors to specify the number of proteins of each subfamily, and comment on whether the datasets are well-balanced.
- Beyond the 5-fold cross validation, how the authors checked for overfitting? This is particularly relevant given that the datasets look quite small. Furthermore, it is not clear if training and test sets were used, and to which one correspond the results reported in Table 3. The results on the test sets are the ones that should be reported.
- The role of different kinds of hydrogen bonds in the classification should be further discussed. Hydrogen bonds involving backbone atoms only, such as in alpha-helices or beta-sheets, are directly related to secondary structure; while sidechain-sidechain and backbone-sidechain hydrogen bonds would be in principle more directly related to loop interactions and preorganization of the active sites. An option would be to perform a drop out analysis building the hydrogen bond graphs considering different kinds of hydrogen bonds, which may reveal the importance of each kind of hydrogen bond in the overall classification. It is expected that backbone-only hydrogen bonds would play a smaller role in explaining ligand-binding preferences across subfamilies.
- As a more general comment, the paper would be of interest to a broader audience if some examples of how the classifier helps to rationalize ligand-binding preferences are provided. For example, would it be possible to identify key hydrogen bond networks or amino acids characteristic of each subfamily?
Author Response
(please see attachment)

Reviewer 3 Report
This manuscript introduces an interesting, seemingly very successful,
graph-theoretic deep learning approach to classifying proteins based
on hydrogen-bonding topology. What is unclear is how classification
would be expanded to an additional class of proteins. Presumably, a
new round of training would be required/.
For this reviewer, a little more detail is required to understand the
results in Tables 3 and 4. Does the accuracy metric mean that all
proteins (from all classes) were placed in classes with the different
methods, and then the percentage in the correct class then reported?
Similarly, how was classification defined for the pairwise (RMSD-like)
metrics in ska and clustalw, reported in Table 4.
For the source code availability, I see that HbondFinder is present on
github as a python program, but HBcompare is entered as a jupyter
notebook, giving the impression that this is not the complete
source. Presumably, after training, one should be able to specify an
arbitrary PDB and the identity of the protein fold returned- how one
accomplishes this is not clear.
Other issues:
- p. 4., immediately below Eq. 2: \tilde{A} should be \hat{A} (the
accent over the A is incorrect).
- Section 3 is labeled ``Experimental Results'': computations rather
than experiments were performed. I would prefer simply ``Results''
- p 10, line 356-357: ``... require on a structural...'' should be
``... require a structural...''
Author Response
(please see attachment)

Round 2
Reviewer 2 Report
The authors properly addressed all my concerns and I am happy to accept the manuscript in its present form.
Author Response
Thanks